# Remote analysis of sputum smears for mycobacterium tuberculosis quantification using digital crowdsourcing

Lara García Delgado[1,2◉], María Postigo[3◉], Daniel Cuadrado[3], Sara Gil-Casanova[1], Álvaro Martínez Martínez[3], María Linares[4,5], Paloma Merino[6], Manuel Gimo[7], Silvia Blanco[7,8], Quique Bassat[7,8,9,10,11], Andrés Santos[1,2], Alberto L. García-Basteiro[8,9,12], María J. Ledesma-Carbayo[1,2‡]*, Miguel Á. Luengo-Oroz[1,2,3‡]

**1** Biomedical Image Technologies, ETSI Telecomunicación, Universidad Politécnica de Madrid, Madrid, Spain, **2** Consorcio de Investigación Biomédica en Red de Bioingeniería, Biomateriales y Nanomedicina (CIBEBBN), Madrid, Spain, **3** Spotlab, Madrid, Spain, **4** Department Biochemistry and Molecular Biology, Pharmacy School, Universidad Complutense de Madrid, Madrid, Spain, **5** Department of Hematology, Hospital 12 Octubre de Madrid, Madrid, Spain, **6** Clinical Microbiology Department, Clinico San Carlos Hospital, Madrid, Spain, **7** Centro de Investigação em Saúde de Manhiça (CISM), Maputo, Mozambique, **8** ISGlobal, Hospital Clínic—Universitat de Barcelona, Barcelona, Spain, **9** ICREA, Barcelona, Spain, **10** Pediatrics Department, Pediatric Infectious Diseases Unit, Hospital Sant Joan de Déu (Universidad de Barcelona), Barcelona, Spain, **11** Consorcio de Investigación Biomédica en Red de Epidemiología y Salud Pública (CIBERESP), Madrid, Spain, **12** Centro de Investigación Biomédica en Red de Enfermedades Infecciosas (CIBERINFEC), Barcelona, Spain

◉ These authors contributed equally to this work.
‡ MJLC and MALO also contributed equally to this work.
* mj.ledesma@upm.es

**Data Availability Statement:** A minimal dataset to reproduce the results is available at: https://doi.org/10.5281/zenodo.6407153.

## Abstract

Worldwide, TB is one of the top 10 causes of death and the leading cause from a single infectious agent. Although the development and roll out of Xpert MTB/RIF has recently become a major breakthrough in the field of TB diagnosis, smear microscopy remains the most widely used method for TB diagnosis, especially in low- and middle-income countries. This research tests the feasibility of a crowdsourced approach to tuberculosis image analysis. In particular, we investigated whether anonymous volunteers with no prior experience would be able to count acid-fast bacilli in digitized images of sputum smears by playing an online game. Following this approach 1790 people identified the acid-fast bacilli present in 60 digitized images, the best overall performance was obtained with a specific number of combined analysis from different players and the performance was evaluated with the F1 score, sensitivity and positive predictive value, reaching values of 0.933, 0.968 and 0.91, respectively.

## Introduction

Tuberculosis (TB) is a leading cause of morbidity and mortality worldwide. Although the development of Xpert MTB/RIF has recently become a major breakthrough, smear micros-copy remains the most widely used method for TB diagnosis, especially in low- and middle-

**Funding:** This research was partially funded by the project H2020-MSCA-RISE-2018 INNOVA4TB (EU), CDTI NEOTEC SNEO-20171197, RTI2018-098682-B-I00 (MCIU/AEI/FEDER, EU) from the Spanish Ministry of Science, Innovation and Universities (MJLC, AS, LGD) and IND2019/TIC-17167 from Comunidad de Madrid (MALO,MJLC). CISM is supported by the Government of Mozambique and the Spanish Agency for International Development (AECID) (MG,SB,QB, AGB). Spotlab provided support in the form of salaries for authors MP, DC, AMM. The funders did not have any role in study design, data collection and analysis, decision to publish, or preparation of the manuscript.

**Competing interests:** MALO, DC, MP, ML, QB, AS and MLJC are founders of Spotlab. MP, DC, AMM are or have been employees of Spotlab during the development of this study. This does not alter our adherence to PLOS ONE policies on sharing data and materials.

income countries [1]. Given its low sensitivity, the World Health Organization (WHO) recommends that three sputum specimens should be examined for each TB presumptive case. Furthermore, in clinical practice, 100 high-power fields need to be examined in order to classify a smear as negative. Acid fast bacilli (AFB) smear reading requires a skilled microscopist and considering the lab workload associated with smear reading, a microscopist can only examine an average of 20–25 smears/day [2]. In addition, smear reading is subject to human error and prone to considerable interobserver variability [3]. Novel approaches, such as automated image analysis through convolutional neural networks, have recently shown promising results performing microscopy tasks as diagnosis of malaria in thick blood smears, tuberculosis in sputum samples, and intestinal parasite eggs in stool samples [4].

Detecting acid fast bacilli in sputum smear samples is a challenge that has been addressed before. In 2008, M.G. Costa et al. [5] published a method based on global adaptive threshold applied to Red and Green color channels of conventional microscopy images, obtaining a sensitivity of 76.7%. In 2018, Kant et al. [6] developed a system based on convolutional neural networks that achieved a recall of 83.8% and a precision of 67.6%. The same year, R.O. Panicker et al. [7] proposed a method that performs detection of tuberculosis bacilli by image binarization and subsequent classification of detected regions using a convolutional neural network obtaining a precision of 78.4%, a recall of 97.1% and a F1 score of 86.8%. To the best of our knowledge, this is the first crowdsourced approach to detect acid fast bacilli in sputum smears samples.

Crowdsourcing methodologies leveraging the contributions of citizen scientists connected via the Internet have shown utility to solve biomedical challenges involving "big data" analysis that cannot be entirely automated [8]. The "gamification" of crowdsourced tasks untaps a resource for scientific research such as biomedical image analysis [9, 10]. In this context, we aimed to evaluate the feasibility of a crowdsourced approach to sputum smear microscopy analysis for the diagnosis of tuberculosis.

## Materials and methods

### The gaming platform

*TuberSpot* (www.tuberspot.org) is an online game for mobile and PC launched on the 24th of March 2015. TuberSpot players score points by identifying correctly *M. tuberculosis bacilli* in digitized sputum slide fields of view (FOVs) with Ziehl-Neelsen stain (Fig 1). Gamers play with several fields images (FOVs) during each game. A backend server shares out randomly the different FOVs to the players in real time.

Once the game starts, the player sees a FOV on the screen and, within a limited time, has to click in the places where bacilli are believed to be present. Once all bacilli are found, players pass to the next level. We have digitally introduced one synthetic bacillus (fake) in each of the negative FOVs, which cannot be distinguished from a normal one, ensuring that enough time is spent in the FOV even if originally there were no bacillus in it and allowing the introduction of negative FOV in the game. At the beginning of the game, there is a short tutorial showing how a bacillus looks like.

### Dataset

The game database consists of 60 digitized FOVs from anonymous samples: 20 images of fields without any bacilli, 20 images with 1–10 bacilli and 20 images with 10–40 bacilli. Digitized smears were provided by the *Centro de Investigação em Saúde de Manhiça* (Mozambique) and Hospital Clínico San Carlos (Spain). The 60 images come from all types of sputum smear examination reports (negative, scanty, +1, +2, +3). Digitalization of the samples was made

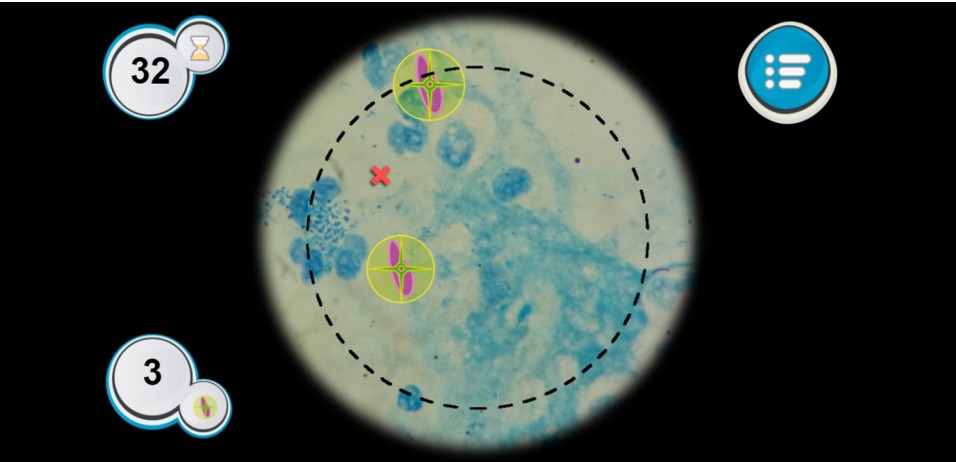

**Fig 1. Screenshot of a "known sample" of TuberSpot game.** In this sample the number of AFB is known, 36 in this case, and the players have to click in the places where bacilli are believed to be. This kind of samples are used to train the players.

with a smartphone (Sony Xperia Z2) attached to the microscope eyepiece by an adapter (Celestron Universal Digiscoping Adapter). A gold standard for each FOV has been determined by three different expert microscopists, reporting the position and number of bacilli.

## Crowdsoucing scheme

Collective detection is defined as the number of bacillus found in a single FOV based on the combination of the gameplays from different players over the same FOV. In order to exploit the redundant information produced by multiple independent players over the same FOV, an algorithm was implemented considering that there is a bacillus in a certain area of the FOV if enough individual players in a larger group have clicked ("voted") in that area of the same FOV [10]. Taking into account that players do not click exactly on the same pixel of the image we applied a clustering strategy. Each point was clustered with the closest neighboring point if the distance between the two points was shorter than the typical size of a bacillus.

To classify a point in the FOV as a bacillus a given number of players must agree: this number is denominated as *Quorum* (Q). Group sizes (GS) from 1 to 30 gameplays and quorums, from 1 to the maximum number of gameplays, were tested to maximize the performance in the whole test dataset.

The performance of the collective detection algorithm has been evaluated for each quorum (Q) and each size of players groups (GS) with respect to the gold standard measuring: the positive predictive value (precision)(1), the sensitivity (recall or true positive rate (TPR)) (2), the F1 score (3) and the specificity (true negative rate TNR) (4). Collective detections with a given Quorum and Group Size were considered true positives (tp) if the positive cluster distance to a gold standard detection is shorter than the typical size of a bacilli. Accordingly, collective detections with distance greater than the typical bacilli size with respect to a reference bacillus are considered as false positives (fp). All the bacilli that were not collectively detected were considered as false negatives (fn). To calculate the true negatives (tn) we measured the number of equivalent bacilli in the area of the field of view were no bacilli were identified by the experts nor by the collective detections. To this end the area of a bacillus and its immediate surroundings (bacillus area) is used to divide the area of the FOV free of bacilli and collective

detections.

$$PPV = \frac{tp}{tp + fp} \tag{1}$$

$$TPR = \frac{tp}{tp + fn} \tag{2}$$

$$F1\ Score = 2 * \frac{PPV * TPR}{PPV + TPR} \tag{3}$$

$$TNR = \frac{tn}{tn + fp} \tag{4}$$

Where:

$tp$ = Bacillus detected by at least Q out of GS.

$fp$ = Point, that is not a bacillus, voted by Q out of GS.

$fn$ = Bacillus that it is not "voted" by Q out of GS.

tn = Number of bacillus equivalent area in a FOV were bacilli are not present and not voted by Q out of GS.

Additionally Cohen's Kappa was computed to assess the agreement between the collective assessment and the reference gold standard.

In Fig 2 there is an example of a field of view in which there are 18 bacilli (red mask), green and red crosses are points where the 20 players clicked on during the gameplay. In the first image, with Q = 2, 2 people out of 20 had to agree clicking in the same area in order to consider that area as a bacillus, with that metric the result would be 18 true positives and 3 false positives. In the second image, with Q = 20, taking into account that group size is 20, all the players of the group had to click on one area to consider it as a bacillus, the result for that experiment is 9 true positives and 9 false positives.

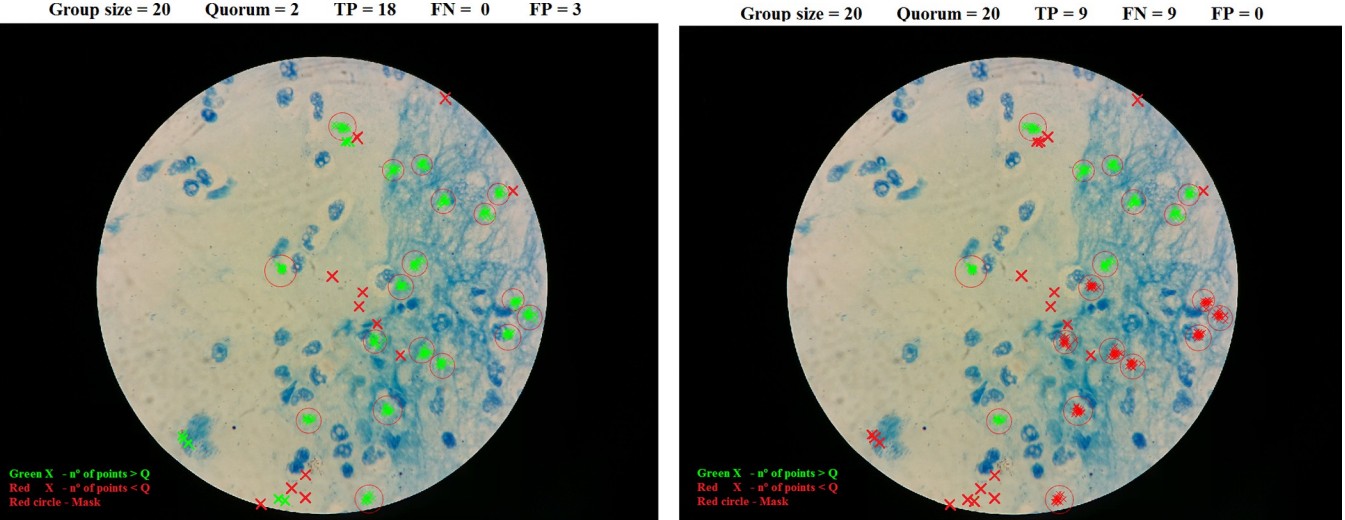

**Fig 2. Examples of two different experiments performed on the same field of view, *left* quorum Q = 2, *right* quorum Q = 20.** Red crosses are clicks that belong to a cluster that have less clicks than the quorum number evaluated in the experiment. Green crosses are clicks that belong to a cluster bigger than the quorum number evaluated. Red circles correspond to confirmed locations of bacilli. TP- true positive, FN- false negative, FP -false positive.

## Experimental setup

The images for this experiment were uploaded to the TuberSpot online platform in April of 2017 and the analysis was performed in February of 2018. During that period, 1790 players analyzed the digitized FOVs reaching a total of 14749 individual FOVs-analysis. The players were not given a specific number of FOV to analyze as a task to complete, every FOV they analysed was considered independently of the number of images they played. The performance of the collective detection was evaluated for group sizes from 1 to 30 gameplays considering a gameplay as a FOV analyzed by a single player. We have analyzed 160 random combinations of gameplays for each group size and each one of the 60 images. Based on that analysis, we have identified the minimum number of players that provide the highest F1 score over all our test dataset. The collective detection based on the optimal size of the groups of players has been evaluated with a confusion matrix with three relevant classes for diagnostic purposes of the FOV: no AFB (or a fake), between 1 and 10 AFB and more than 10 AFB.

## Results and discussion

The best overall performance was obtained for a mean F1 score of 0.933, a sensitivity of 0.968, a positive predictive value of 0.916, specificity of 0.998 and a kappa statistic of 0.927 for the combination of 29 gameplays and quorum 18 in comparison to expert microscopists. A very competitive result considering a smaller group size is achieved for group size 8 and quorum 5, for this combination the F1 score is 0.917, with a sensitivity of 0.963, a positive predictive value of 0.893, specificity of 0.998 and a kappa statistic of 0.905 (Fig 3). According to the guideline for sputum examination for tuberculosis by direct microscopy in low income countries proposed by the IUATLD [2], there are three different classifications regarding the AFB counts per FOV: no AFB, 1–10 AFB and >10 AFB. Based on that classification per FOV and the quantity of FOV per smear sample with a specific classification, the severity of the disease is determined. In order to test our methodology following this guideline, we classified the FOV analyzed for the combination of 8 gameplays with quorum of 5 (Table 1).

This analysis shows that it is possible to identify and count *M. tuberculosis AFB* in digitized sputum smears based on the data produced by a number of non-expert on-line volunteers playing a video game over the same FOVs. Results from the collective detection with high accuracy for a group size of 29 players and a quorum of 18 against expert microscopists as gold

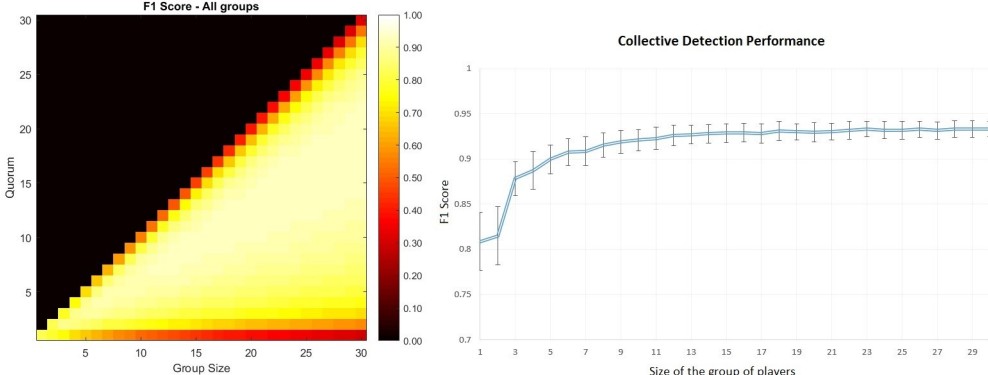

**Fig 3. Mean F1 score and collective detection performance.** Left. Mean F1 score over the whole image dataset depending on the size of the group of players and quorum. Each combination of group size and quorum was tested 160 times through random selection of gameplays of each image. Right. Mean and standard deviation of overall F1 score corresponding to the best quorum for each group size.

**Table 1. Confusion matrix of the reference number of *M. tuberculosis* per image (rows) vs number of *M. tuberculosis* found by players (columns).** Results shown in this matrix were achieved for a collective detection made by groups of 8 people with a quorum of 5.160 random groups of 8 players for each one of the FOVs. Numbers in the confusion matrix represent the percentage of the gameplays analyzed that were classified as a negative FOV (with zero bacillus), as a FOV with a fake bacillus, as a FOV with 1 to 10 bacilli or as a FOV with more than 10 bacilli.

| Collective Detection Vs. Gold standard | Negative | | 1–10 AFB | AFB>10 |
|---|---|---|---|---|
| | Negative | Fakes | | |
| Negative (with fake) | 0.03% | 93.54% | 6.43% | 0% |
| 1–10 AFB | 0.03% | | 99.63% | 0.34% |
| AFB>10 | 0% | | 16.31% | 83.69% |

standard, and a very competitive result for a smaller number of gameplays, group size 8 and quorum 5. According to the TB reporting IUATLD guideline it is necessary to count the *AFB* present on many FOVs in order to report the grade of infection of the patient. Therefore, further experiments with our system should be done to evaluate the performance following the entire diagnostic protocol for which specificity and sensitivity should be reported. Broadly, this research has defined the design criteria for a real-time remote analysis system for performing routine *M. tuberculosis* quantification that could be applied in endemic settings, characterized by a lack of expert microscopists [10].

Current trends and recommendations for TB diagnostics are shifting from microscopy confirmation towards molecular methods such as Xpert, Turenat etc. . . However, according to the latest Global Tuberculosis Report by WHO [11] rapid molecular tests were only used in 33% of the total people newly diagnosed in 2020 due to lack of accessibility. In this context, the proposed concept could still have potential in those settings where AFB is the main diagnostic tool in place and could overcome some of the limitations associated with AFB reading (mainly single operator dependent reading, lack of training of readers in many settings and time consumption in already overburdened lab technologists). As limitations, our solution could not be useful for TB diagnosis in vulnerable groups such as people living with HIV and children given the low sensitivity of sputum smear tests in these groups.

On the other hand, in remote high burden settings, mobile phone and SMS-based technologies are emergent enabling tools to increase the rate of case detection by improving the efficacy of specimen collection and reporting results of acid-fast bacilli (AFB) microscopy, or to improve reporting and management for rapid diagnostic testing of HIV and malaria [12, 13]. Such system could be operationalized obtaining images from a mobile microscope system [14, 15] and the distribution of images through the internet via the mobile network, provided that there is connectivity which might not be the case in remote and rural environments. Moreover, it would be necessary to include a step to identify color blindness, or develop alternative representations for wider accessibility, as the analysis could be altered if this disease is considered.

This concept has been piloted in relevant operational conditions including the acquisition of the data through a mobile phone adapted to a conventional microscope, data being sent to the game through the mobile phone data connection and receiving the assessment from the online game of the uploaded FOV after a given time. The system was tested during a specific campaign taking place within a few days at Centro de Investigação em Saúde de Manhiça (CISM) in Mozambique showing an assessment turnaround per case of 15–30 min after the image was uploaded into the game, with around 100 simultaneous players.

This technological set up has the potential to be converted into a remote diagnosis platform connecting volunteer players, microscopists and specifically trained remote digital workers (micro-workers) that could get an incentive for their assessment generating flexible labor posts

in a similar scheme to Amazon Mechanical Turk. Players would be rated on their level of expertise and their performance within the game. The representative FOVs (negative FOVs, FOV with big number of bacilli and FOV with a medium number of bacilli) of crowdsourced samples FOVs assessed by less experienced players would be sent to expert microscopists for diagnosis confirmation integrating the results of the crowdsourcing system.

Another possible way to exploit the diagnostic utility of this technology would be to prioritize reading for human experts based on the sample classification performed by the players. The samples in which the players identified the highest number of bacilli would be sent first to expert microscopists to obtain a confirmation as soon as possible using the crowdsourcing system report and the collective detections on the images to facilitate the process. Following, the cases with less detected bacilli, and specially the challenging cases with few or no identified bacilli, would be sent to experts to obtain an additional detailed reading and confirmation.

For both scenarios, specific studies would be necessary to assess the speed with which digital workers and players respond as well as the stability of the network that is required to ensure that diagnosis would arrive rapidly to remote areas. Certification of the system and the workflows should follow these studies before they could be integrated in the real clinical settings.

Furthermore, there is a relevant potential of incorporating this type of strategies as an evaluation of External Quality Assurance schemes, validating the trainee's performance in a more friendly way for a certain period of time, increasing, accordingly to the complexity of the images, the length of the training and the digital game levels. Although fluorescence microscopy has better accuracy, in the settings where this technology would be useful this technique is not widely available.

Comparing the results obtained through this crowdsourced approach to the ones obtained with deep learning techniques [4–7], we believe crowdsourcing methodologies can provide added value to traditional image-based diagnostics. Additionally, as recently published for helminthiasis samples [16], this type of systems can produce expert level labelled data that can be used to train artificial intelligence systems and contribute to the definition of new digital diagnostic methodologies that combine artificial intelligence systems [17, 18] and human intelligence.

Lastly, this approach might also be very relevant for educational purposes and a powerful tool for advocacy, especially among young people. This has been proven during the past years, more than 5000 children and young people in Spain have participated in workshops with videogames for global health.

## Acknowledgments

We thank the thousands of TuberSpot players for their passion and their games. ISGlobal is a member of the CERCA Programme, Generalitat de Catalunya (http://cerca.cat/en/suma/).

## Author Contributions

**Conceptualization:** Paloma Merino, María J. Ledesma-Carbayo, Miguel Á. Luengo-Oroz.

**Data curation:** Daniel Cuadrado, María Linares.

**Formal analysis:** Lara García Delgado.

**Funding acquisition:** Sara Gil-Casanova, Andrés Santos, Miguel Á. Luengo-Oroz.

**Investigation:** Lara García Delgado, María Postigo, Sara Gil-Casanova, María Linares, Paloma Merino, Alberto L. García-Basteiro, Miguel Á. Luengo-Oroz.

**Methodology:** Lara García Delgado, María Postigo, María Linares, Quique Bassat, Andrés Santos, Alberto L. García-Basteiro, María J. Ledesma-Carbayo, Miguel Á. Luengo-Oroz.

**Project administration:** Daniel Cuadrado, Sara Gil-Casanova, María J. Ledesma-Carbayo, Miguel Á. Luengo-Oroz.

**Resources:** Paloma Merino, Manuel Gimo, Silvia Blanco, Quique Bassat, Alberto L. García-Basteiro, María J. Ledesma-Carbayo, Miguel Á. Luengo-Oroz.

**Software:** Daniel Cuadrado, Álvaro Martínez Martínez, Miguel Á. Luengo-Oroz.

**Supervision:** Quique Bassat, María J. Ledesma-Carbayo, Miguel Á. Luengo-Oroz.

**Validation:** Daniel Cuadrado, Miguel Á. Luengo-Oroz.

**Visualization:** Daniel Cuadrado, Paloma Merino.

**Writing – original draft:** Lara García Delgado, María Postigo.

**Writing – review & editing:** María Postigo, Quique Bassat, Andrés Santos, Alberto L. García-Basteiro, María J. Ledesma-Carbayo, Miguel Á. Luengo-Oroz.

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
