## [Decision Letter · Decision Letter 0]

24 Feb 2021

PONE-D-21-00739

Remote analysis of Sputum Smears for Mycobacterium Tuberculosis Quantification using Digital Crowdsourcing

PLOS ONE

Dear Dr. Delgado,

Thank you for submitting your manuscript to PLOS ONE. After careful consideration, we feel that it has merit but does not fully meet PLOS ONE’s publication criteria as it currently stands. Therefore, we invite you to submit a revised version of the manuscript that addresses the points raised during the review process.

Please submit your revised manuscript. If you will need significantly more time to complete your revisions, please reply to this message or contact the journal office at plosone@plos.org. Please include the following items when submitting your revised manuscript:

We look forward to receiving your revised manuscript.

Kind regards,

Frederick Quinn

Academic Editor

PLOS ONE

Journal Requirements:

2.In your Data Availability statement, you have not specified where the minimal data set underlying the results described in your manuscript can be found. PLOS defines a study's minimal data set as the underlying data used to reach the conclusions drawn in the manuscript and any additional data required to replicate the reported study findings in their entirety. All PLOS journals require that the minimal data set be made fully available. For more information about our data policy, please see http://journals.plos.org/plosone/s/data-availability.

3.Thank you for stating the following in the Acknowledgments Section of your manuscript:

"This research was partially funded by the project H2020-MSCA-RISE-2018 INNOVA4TB (EU), CDTI NEOTEC

SNEO-20171197 and TEC2015-66978-R (MINECO/FEDER UE) from the Ministry of Science, Innovation and

Universities."

4.Thank you for stating the following in the Competing Interests section:

We note that one or more of the authors are employed by a commercial company:Spotlab, Madrid

Reviewers' comments:

Reviewer's Responses to Questions

**Comments to the Author**

1. Is the manuscript technically sound, and do the data support the conclusions?

Reviewer #1: Yes

Reviewer #2: Partly

Reviewer #3: No

2. Has the statistical analysis been performed appropriately and rigorously? 

Reviewer #1: I Don't Know

Reviewer #2: Yes

Reviewer #3: No

3. Have the authors made all data underlying the findings in their manuscript fully available?

Reviewer #1: Yes

Reviewer #2: No

Reviewer #3: Yes

4. Is the manuscript presented in an intelligible fashion and written in standard English?

Reviewer #1: Yes

Reviewer #2: Yes

Reviewer #3: No

5. Review Comments to the Author

Reviewer #1: Authors analyze the feasibility of a crowdsourced approach to tuberculosis image analysis that presents interesting results.

Although the crowdsourced approach nor the application in health problems are not new, the specific application is interesting and the obtained results suggest a possible use in tuberculosis diagnosis.

It is convenient to enhance the quality of the images.

The used methodology should be more descriptive.

Reviewer #2: Major comments:

The manuscript " Remote analysis of Sputum Smears for Mycobacterium Tuberculosis Quantification using Digital Crowdsourcing" reports the feasibility of a crowdsourced approach to sputum smear microscopic analysis for the diagnosis of tuberculosis. The authors investigated whether volunteers (game players) would be able to count acid-fast bacilli in digitized images of sputum smears with an online game. Against expert microscopists as gold standard, the F1 score, sensitivity and positive predictive of volunteers reached to 0.933, 0.968 and 0.916, respectively. In conclusion, the authors expected that their system, in which acid-fast bacilli (AFB) microscopic images obtained by a mobile phone were distributed through the mobile network, could be an effective tool as a remote diagnosis platform or an educational tool. They also believed that crowdsourced methodologies employed in this study could produce practical data for other diagnostic methods and development of artificial intelligence systems.

Technically, providing AFB smear images obtained by mobile phone to expert healthcare professionals in remote area could be a useful tool for high burden areas without adequate medical systems. It is also very important to show the diagnostic ability of non-professional volunteers could be comparable using a crowdsourced approach with well verified criteria. As authors mentioned, this approach could product constructive data for developing not only human intelligence but artificial intelligence systems.

As for TB diagnosis, methods processing thousands of AFB smear samples would not be practical. Basically, it is hard to obtain appropriate sputum sample, especially from children, extrapulmonary tuberculosis and disseminated TB. The latter two are often observed people living with HIV. Children and people living with HIV are the most vulnerable population to TB. Therefore, the World Health Organization calls for the development of a rapid and non-sputum tests capable of detecting TB at the point-of-care (POC), because sputum smear test has low diagnostic sensitivity in children, extrapulmonary TB, and people living with HIV.

Developing a system for resource-limited area, which is operated by a crowdsourced concept via mobile phone network, has various utilities not only for image diagnostic purposes but other fields. However, for reasons mentioned above, developing a system for AFB smear diagnosis would not efficiently contribute to the incremental diagnostic yield of TB.

Minor points

tp= Bacillus detected by at least Q out of SG players.

fp = Point, that is not a bacillus, voted by Q out of SG players.

fn= Bacillus that it is not “voted” by Q out of SG players.

No explanation for abbreviation “SG”. What is SG stand for?

Reviewer #3: Microscopy will continue to play a key role in diagnosis and and treatment monitoring of tuberculosis. Efforts to improve performance are lauded such as the work that is being presented here. However, they are several concerns.

Major

1. The authors have only reported precision ( F1 score), sensitivity and PPV. Use of F1 score is limited in that it does not take into account true negative ( specificity). It is advisable to use either Mathews correlation or Kappa statistic when assessing binary classifiers. The specificity should also be reported.

I would interested to see these analysis before making any decision on this submission

2.It is not immediately clear how crowdsourced tasks would be used in clear setting. Does it mean that reading of smears could in future be done through crowd sourced reading (gamification?)

Minor

1. The manuscript does not flow well . The first paragraph of results and discussion seems to seems to fit better under methodology

2. There are no line numbers which makes it difficult to pint out exact areas/ sentences where comment is directed

6. PLOS authors have the option to publish the peer review history of their article (what does this mean?). If published, this will include your full peer review and any attached files.

Reviewer #1: No

Reviewer #2: No

Reviewer #3: No

---

## [Author Response · Author response to Decision Letter 0]

1 Apr 2022

Response to the reviewers

Reviewer #1: Authors analyze the feasibility of a crowdsourced approach to tuberculosis image analysis that presents interesting results.

Although the crowdsourced approach nor the application in health problems are not new, the specific application is interesting and the obtained results suggest a possible use in tuberculosis diagnosis.

Thank you for your comment, indeed this technology demonstrates that through visual search of people without specific knowledge it is possible to detect acid-fast bacilli in sputum samples. In addition, as described in the article, another application of this technology would be the generation of samples for training expert systems to analyse the images automatically (page 7, lines 236-238).

R1.1 It is convenient to enhance the quality of the images.

We appreciate your suggestion; we have revised the images, and we now append higher quality ones (original TIFF format) which allow for easier reading/interpretation. 

R1.1 The used methodology should be more descriptive.

Thank you for pointing this out. We agree we should improve the description of part of the methods of our study. Therefore, we have restructured the methodology and we have extended and clarified some points (Materials and Methods section, pages 2-4). In addition, we have included a paragraph explaining how the clustering strategy was implemented (page 3, lines 99-102).

Reviewer #2: Major comments:

R2.1 The manuscript "Remote analysis of Sputum Smears for Mycobacterium Tuberculosis Quantification using Digital Crowdsourcing" reports the feasibility of a crowdsourced approach to sputum smear microscopic analysis for the diagnosis of tuberculosis. The authors investigated whether volunteers (game players) would be able to count acid-fast bacilli in digitized images of sputum smears with an online game. Against expert microscopists as gold standard, the F1 score, sensitivity and positive predictive of volunteers reached to 0.933, 0.968 and 0.916, respectively. In conclusion, the authors expected that their system, in which acid-fast bacilli (AFB) microscopic images obtained by a mobile phone were distributed through the mobile network, could be an effective tool as a remote diagnosis platform or an educational tool. They also believed that crowdsourced methodologies employed in this study could produce practical data for other diagnostic methods and development of artificial intelligence systems.

Technically, providing AFB smear images obtained by mobile phone to expert healthcare professionals in remote area could be a useful tool for high burden areas without adequate medical systems. It is also very important to show the diagnostic ability of non-professional volunteers could be comparable using a crowdsourced approach with well verified criteria. As authors mentioned, this approach could product constructive data for developing not only human intelligence but artificial intelligence systems.

Thank you for your comment; we appreciate the time invested in reviewing our work. As you point out, we believe that this technology could be useful to support the diagnosis of tuberculosis or other diseases in places where access to a quality health system is limited. In addition, bearing in mind that medical imaging for training systems based on artificial intelligence is often limited, it could also provide added value for the sector.

We agree with the reviewer that to make the comparison of the non-expert volunteers using a crowdsource approach should be based on verified criteria. Following this quality principle, in this work three experts annotated the images used as gold standard (page 3 lines 86-87). Furthermore, the strategy applied for clustering was that the distance between clicks should be less than or equal to the size of a bacillus, to ensure that both players were pointing to the same object in the image. Taking into account the strategy employed, we believe that the system is robust and could be used for the use cases described.

R2.2 As for TB diagnosis, methods processing thousands of AFB smear samples would not be practical. Basically, it is hard to obtain appropriate sputum sample, especially from children, extrapulmonary tuberculosis and disseminated TB. The latter two are often observed people living with HIV. Children and people living with HIV are the most vulnerable population to TB. Therefore, the World Health Organization calls for the development of a rapid and non-sputum tests capable of detecting TB at the point-of-care (POC), because sputum smear test has low diagnostic sensitivity in children, extrapulmonary TB, and people living with HIV. Developing a system for resource-limited area, which is operated by a crowdsourced concept via mobile phone network, has various utilities not only for image diagnostic purposes but other fields. However, for reasons mentioned above, developing a system for AFB smear diagnosis would not efficiently contribute to the incremental diagnostic yield of TB.

Thank you for your suggestion. We agree that a potential implementation of this tool could be contentious and would be deemed by some as “unnecessary” given that TB laboratory confirmation is shifting to molecular methods (Xpert, LPAs, Truenat (MolBIO), etcetera) as compared to classical microbiological methods. 

We also agree that there is a need for better tools for HIV associated TB ad pediatric TB. However, this study does not go against the development of the much-needed non-sputum rapid tests. However, as of today, a majority of patients with presumptive TB receive a smear microscopy exam as frontline test and the use of rapid molecular tests remains far too limited. In 2020, a WHO-recommended rapid molecular test was used as the initial diagnostic test for only 1.9 million (33%) of the 5.8 million people newly diagnosed with TB (Global Tuberculosis Report, 2021). If this situation remains for the years to come (as acknowledged by reviewer 3), we believe our proposed concept could still have potential in some of the settings where AFB is the main diagnostic tool in place. It is indeed an innovative solution that can overcome some of the limitations associated with AFB reading (mainly single operator dependent reading, lack of training of readers in many settings and time consumption in already overburdened lab technologists) in those places where they still use it because there are not enough resources or capacity for doing molecular testing.

We agree with the reviewer that our concept would not contribute to the incremental diagnostic yield of TB, although we believe it would potentially have an impact on health system effectiveness. 

We have modified the discussion limitations section to include the previous comments (page 5-6 lines 188-196) including the limitation of the utility of these tools for vulnerable groups such as people living with HIV and children.

Minor points

tp= Bacillus detected by at least Q out of SG players.

tp = Point, that is not a bacillus, voted by Q out of SG players.

fn= Bacillus that it is not “voted” by Q out of SG players.

No explanation for abbreviation “SG”. What is SG stand for?

We apologize for the confusion; GS stands for group size, as described in line 104 (page 3). We have corrected the error in the new version of the article.

Reviewer #3: 

Microscopy will continue to play a key role in diagnosis and treatment monitoring of tuberculosis. Efforts to improve performance are lauded such as the work that is being presented here. However, they are several concerns.

Major

R3.1. The authors have only reported precision ( F1 score), sensitivity and PPV. Use of F1 score is limited in that it does not take into account true negative (specificity). It is advisable to use either Mathews correlation or Kappa statistic when assessing binary classifiers. The specificity should also be reported.

I would interested to see these analysis before making any decision on this submission

We are grateful for the suggestion to include specificity as a measure of the performance of this technology. We agree that it provides more evidence of adequate performance; therefore, we have included it in the new version of the article.

The calculation of true positive bacilli detections, false positives, false negatives and true negatives based on the collective detection is now more clearly explained in the methods section analysis (pages 3 and 4 lines 107 to 124). Specificity and Kappa are included in the statistical analysis of performance of the collective detections. 

In the results section specificity and kappa are now reported (page 4 lines 153-157). Very good agreement and high specificity is shown for the selected Quorum (Q=18) and Size Group (SG=29) as well as for Quorum (Q=5) and Group Size (GS=8). 

We have also added a comment in the discussion emphasizing that the specificity and sensitivity for the entire diagnostic protocol following the IUATLD guideline should be further investigated and reported (page 5, line 184). 

R3.2.It is not immediately clear how crowdsourced tasks would be used in clear setting. Does it mean that reading of smears could in future be done through crowd sourced reading (gamification?)

We are grateful for your comment and have therefore expanded the part of the discussion to include possible workflows of how this methodology could be used to support TB diagnosis (page 6 lines 213-229). In brief, we agree that it is still unclear on the implementation hurdles of gamification solutions in high TB burden low income country settings. However, this is slightly beyond the scope of this pilot study. However, we envisage an implementation study as the next step where we could measure (beyond diagnostic performance indicators), all process and feasibility aspects that could provide valuable information for a potentially much broader use.

Minor

1. The manuscript does not flow well. The first paragraph of results and discussion seems to fit better under methodology

We agree with this comment and we have incorporated your suggestion revising the structure of the methodology and moving specifically the first paragraph of the results to the methodology as experimental setup. 

Additionally the paper has been revised for English correctness. 

2. There are no line numbers which makes it difficult to pint out exact areas/ sentences where comment is directed.

We apologize for not having incorporated the line numbers to facilitate the revision in the original version, we have incorporated them in the new uploaded version.

---

## [Decision Letter · Decision Letter 1]

3 May 2022

Remote analysis of Sputum Smears for Mycobacterium Tuberculosis Quantification using Digital Crowdsourcing

PONE-D-21-00739R1

Dear Dr. Ledesma-Carbayo,

We’re pleased to inform you that your manuscript has been judged scientifically suitable for publication and will be formally accepted for publication once it meets all outstanding technical requirements.

Kind regards,

Frederick Quinn

Academic Editor

PLOS ONE

Additional Editor Comments (optional):

Reviewers' comments:

Reviewer's Responses to Questions

**Comments to the Author**

1. If the authors have adequately addressed your comments raised in a previous round of review and you feel that this manuscript is now acceptable for publication, you may indicate that here to bypass the “Comments to the Author” section, enter your conflict of interest statement in the “Confidential to Editor” section, and submit your "Accept" recommendation.

Reviewer #1: All comments have been addressed

2. Is the manuscript technically sound, and do the data support the conclusions?

Reviewer #1: Yes

3. Has the statistical analysis been performed appropriately and rigorously? 

Reviewer #1: N/A

4. Have the authors made all data underlying the findings in their manuscript fully available?

Reviewer #1: No

5. Is the manuscript presented in an intelligible fashion and written in standard English?

Reviewer #1: Yes

6. Review Comments to the Author

Reviewer #1: It seems to me that authors have fulfilled the reviewers' comments. The paper can be published in its current form.

7. PLOS authors have the option to publish the peer review history of their article (what does this mean?). If published, this will include your full peer review and any attached files.

Reviewer #1: No

---

## [Editor Report · Acceptance letter]

12 May 2022

PONE-D-21-00739R1 

Remote analysis of Sputum Smears for Mycobacterium Tuberculosis Quantification using Digital Crowdsourcing 

Dear Dr. Ledesma-Carbayo:

I'm pleased to inform you that your manuscript has been deemed suitable for publication in PLOS ONE. Congratulations! Your manuscript is now with our production department. 

Kind regards, 

on behalf of

Dr. Frederick Quinn 

Academic Editor

PLOS ONE